# The Impact of the Low-Carbon Energy Concept and Green Transition on Corporate Behaviour—A Perspective Based on a Contagion Model

Shuran Wen [1], Wei Cui [1,*] and Guiying Wei [2]

[1] School of Economics and Management, China University of Geosciences (Beijing), Beijing 100083, China
[2] School of Economics and Management, University of Science and Technology Beijing, Beijing 100083, China
* Correspondence: cuiw@cugb.edu.cn

**Abstract:** With the globalisation of the economy and the increasing interconnectedness of individuals in the financial markets, companies implementing high energy consumption strategies have become more widespread due to the "herding effect" as they become more closely linked for development. In the context of carbon neutrality, the issue of how to reduce the spread of high energy consumption strategies and the issue surrounding the governance of corporate emissions have become a focus of research. This paper uses the improved SEIJRS infectious disease model to investigate the phenomenon of corporate high energy strategy infection, combined with optimal control theory, to provide a reference for governments and regulators to develop reasonable optimal prevention and control strategies.

**Keywords:** dissemination of beliefs; herd effect; optimum control; infectious disease model

## 1. Introduction

For a long time, there have been significant impacts of climate change on health, agriculture, economy, conflict, migration, etc. [1,2]. Reducing greenhouse gas emissions has become the consensus of the international community. The year 2020 was the deadline for submitting net-zero emission targets [3], and 53 countries already achieved carbon peaking by 2020 [4]. Economic recovery through green and low-carbon development has become the general consensus of the international community. In the last two years, the normal functioning of the economy and society in various countries has suffered a serious blow due to the impact of COVID-19 [5]. In addition, the world's international situation has become increasingly complex, with the trade war between the US and China, the Russia–Ukraine conflict and a volatile international situation that has significantly depressed the global economy. The sudden outbreak of the epidemic and the overlap of international conflicts have caused climate change, which should have been receiving constant attention, to be ignored and the corporate low-carbon transition to be put on hold, but also created some potential opportunities [6]. Some companies have started to adopt high energy consumption strategies in order to maintain their economic level, or due to factors such as media attention. Green technology reserves financial slack and many other factors all have an effect on corporate carbon strategies [7]. The spread of high energy consumption strategies implemented by energy enterprises in the market has a high degree of similarity to infectious diseases. When the concept of green and low-carbon strategies among enterprises is not solid, the government needs to control and govern in a timely manner to prevent the cross-infection of high energy consumption concepts, resulting in environmental pollution [8]. So far, governments around the world have realised the importance of transforming energy-intensive enterprises, and governments have begun to adopt policies of varying degrees [9]. Due to this background, in this paper,

we propose a new SEIJRS epidemic model that takes into account the fact that energy-intensive enterprises can be infected multiple times, including latent and quarantine states, and considers the ability of enterprises to heal themselves in the latent state and the intensity of government control in the quarantine state as dynamic variables. This paper uses the improved SEIJRS infectious disease model to analyse the spread of green beliefs among different types of companies at different stages in a complex network and combines it with optimal control theory to provide a reference for governments and managers to formulate reasonable and optimal prevention and control strategies.

The structure of this paper is as follows: In Section 2, the literature review is presented; this paper analyses the significance of the contagion model for the spread of green and low-carbon beliefs by analysing China and companies under the carbon neutrality target, in conjunction with the herd effect. In Section 3, the methodology, model assumptions, construction and analysis are carried out. In Section 4, this paper solves the optimal control problem. In Section 5, numerical simulations are carried out to obtain the dynamics of the various results for different situations and to plot the images in order to draw conclusions about the optimal control. Conclusions and limitations are given in Section 6, which is followed by conclusions in Sections 3–5.

## 2. Literature Review

### 2.1. China under the Goal of Carbon Neutrality

Until 2019, national carbon dioxide ($CO_2$) emissions were 10.3 Gt ($\pm 13\%$, confidence interval (CI) = 90%) and per capital $CO_2$ emissions were approximately 7.4 t [10]. Due to rapid economic development and urbanisation, China is now the world's largest emitter of $CO_2$, already accounting for 28% of global $CO_2$ emissions in 2019 [11]. In September 2020, China set the goal of "striving to peak $CO_2$ emissions by 2030 and working towards carbon neutrality by 2060". Achieving the "double carbon" target places higher demands on China's energy consumption intensity and total control. As the world's largest developing country, China's economic growth and carbon emissions are still not fully decoupled, and the drivers of economic growth are still highly dependent on energy-intensive industries. Of these, China's $CO_2$ emissions are mainly attributed to its fossil fuel energy use and manufacturing-based industrial system [12]. At present, the main development mode of China's manufacturing industry is high energy consumption and low efficiency. The development of the manufacturing industry ignores the coordinated development of energy consumption and the environment [13]. Therefore, the proper handling of the four pairs of synergies between development and emissions reduction, overall and local, long-term and short-term, and government and market has a positive effect on promoting economic development and achieving the "double carbon" target. For now, China has launched a low-carbon city pilot project to create green, low-carbon cities by limiting carbon emissions through the actions of businesses and individuals [14].

China needs to improve its green and low-carbon policy and market system and increase investment in the development of renewable energy, as well as encourage more social capital to invest in ecological civilisation, green and low-carbon development, climate governance and other areas. It is important to develop an emissions reduction programme that is synergistic with economic and social development goals, and to explore the path of high-quality transformational development in China under the constraints of carbon neutrality targets.

### 2.2. Enterprises under the Goal of Carbon Neutrality

A low-carbon economy is the ideal model for economic development, where GDP revenues increase and energy consumption decreases. A lower carbon footprint is an economic model that is good for people and good for the planet [15,16]. Hafeznia et al. [17] argue that a low-carbon economy is a form of economy that falls under the category of sustainable development and that a low-carbon development model is a specific pathway to a low-carbon economy. Business is a major player, contributor and leader in achieving a

low-carbon economy. Recently, some scholars have proposed that enterprise heterogeneity should be fully considered to develop more targeted energy saving and emissions reduction policies [18,19]. The importance of corporate emissions reduction is becoming increasingly apparent, and research by experts and scholars on corporate low-carbon strategies is heating up, with some valuable academic results achieved. Pinkse [20], Kim [21], and Zhou and Wen [22] collected and sorted out the low-carbon practices of enterprises and initially established the conceptual framework of low-carbon practices of enterprises. Creating a social environment for high-quality development throughout society requires promoting entrepreneurship, craftsmanship and green consumption awareness. Companies in all countries are now actively addressing climate change and are developing strategies with carbon neutrality as a strategic goal. In order to achieve "peak carbon and carbon neutrality", companies are facing a continuous transformation and upgrade of their industrial and energy structures.

Some companies have implemented high energy consumption strategies, either voluntarily or by force, due to restrictions on their business operations and increased costs of declining revenues. With the increasing intensity of linkages between the various energy companies, individual companies have resorted to high energy consumption strategies for short-term gains in the face of market turbulence. This can have an impact on other companies that use green and environmentally friendly strategies. In order to reduce the occurrence of this phenomenon, the ethics of companies themselves, government regulation and decision making by senior leaders play a key role [23]. Zhu and Sarkis [24] found through their analysis that external constraints such as government regulation and market norms have a positive effect on corporate low-carbon practices. As the corporate low-carbon operational strategists, senior leaders have the important task of identifying the organisational environment, identifying organisational capabilities and undertaking organisational strategic planning and change [25]. At the same time, government management departments and relevant management agencies should also start from the enterprise management level to regulate and standardise the enterprise, as well as the whole. Government management is an important tool to improve the green development of industry and the sound industrial low-carbon economic system. Currently, the Chinese government is constantly refining and adjusting its green innovation strategy. Mi, Weisong et al. extended the analysis of "incremental" spatial and temporal characteristics based on "total" green innovation, providing a basis for the government to adjust green innovation policies [26].

### 2.3. Herd Effect

The "herd effect" is essentially the herding behaviour of decision makers who ignore private information and follow the masses in an environment of information uncertainty [27]. Keynes [28] was the first to point out the herding effect in financial markets, arguing that it was more costly for investors to collect and process information about companies. With less information transparency, decision makers are more likely to be "unsure" and end up blindly following the actions of others, which contributes to the herding effect. Allen [29] defines the high degree of correlation prevalent between markets as the interdependent effect of markets. In the financial markets, the same types of sewage companies have the need to obtain information from peer companies and learn from them in terms of decision making. Welch [30] proposed a herding theory based on an information waterfall, where managers have different access to information in the market, resulting in different decisions being made. Different types of companies therefore have different attitudes and resilience to high energy consumption strategies. Manski [31] systematically attributed the interaction between focal and peer individuals to the "peer effects" hypothesis, stating that peer effects are the result of the interaction between decision maker preferences, cognitive abilities and external environmental constraints. As a result, the "herding effect" in financial markets, where individual companies take high energy consumption decisions and profit, has increased market volatility. This could lead to copycat behaviour by other companies of the same types of emissions and lead to increased environmental pollution,

which is not in line with the context of China's green development and the goal of carbon neutrality. The external environment should restrain this phenomenon at this time, i.e., the government administration should take appropriate control strategies to prevent the herd effect from having further influence.

*2.4. The Significance of the Infectious Disease Model for the Spread of Green Low-Carbon Beliefs*

Cha and Sekyung [32] use a multivariate VAR model to study the risk contagion from the US stock market to emerging country stock markets. However, the VAR approach to calculating the risk of a financial asset or portfolio is based on statistical analysis of past return characteristics to predict the volatility and correlation of its price and, thus, estimate the maximum possible loss. The analysis is not comprehensive as it is based solely on the objective probability that a risk may cause a loss and focuses only on the statistical characteristics of the risk. Bae [33] uses a polynomial logit model to estimate the probability of financial crisis contagion and finds a strong spillover effect between financial markets in Latin America. However, it only warns of speculative shock currency crises on the basis of a few key financial assets or economic indicators, such as interest rates and exchange rates, and is of little significance to the study of the diffusion of low-carbon ideas.

Academic research on the transmission of beliefs and ideas is currently relatively sparse. The process of the transmission of low-carbon perceptions and financial risks has similarities to the process of the transmission of infectious diseases. Firstly, from an individual point of view, energy-consuming enterprises themselves have different green concepts and different abilities to resist the idea of high energy consumption, and some of them are susceptible to infection due to the lack of a strong belief in low-carbon consumption. The states of energy-consuming companies when impacted by energy-intensive ideas are similar to the susceptible, latent, infected, recovered and healthy state of individuals in the epidemic model, which is the basis for studying the spread of energy-intensive strategies using the epidemic model. Secondly, due to the dissemination of information, there is a circulation of ideas surrounding high energy consumption between companies. Once a company adopts a high energy consumption strategy, it is passed on to other companies, and if it is not effectively controlled, the perception spreads, much like the process of the contact transmission of an infectious disease. Thirdly, the perception of high energy consumption is persistent. It can be seen that perceptions of high energy consumption have characteristics similar to the individual variability and linkage of infectious diseases and the spread and diffusion of perceptions.

Prevention and control strategies need to be rationalised in order to prevent cross-contamination and increased environmental pollution from the actions of companies implementing highly polluting decisions [34]. The SIR infectious disease model, first proposed using mathematical theory as a framework, includes three categories of people: susceptible, infected and recovered. Today, an increasing number of scholars are using improved classical contagion models for financial risk, belief transmission and other issues. King [35] argues that the ability of information and resources to be transmitted and allocated across different financial markets can create a contagion effect. May [36] and others point out that finance and ecosystems are interconnected and that the process of financial risk transmission is very similar to that of epidemics such as SARS and the New Guinea epidemic. As a result, contagion dynamics models (SI, SIR, ISR models) have gradually been introduced by scholars from various countries into the field of economics and finance to study financial crisis contagion [37], bank risk contagion, investor sentiment propagation [38], stock market crisis propagation and financial system stability [39], etc., although the realism of the model needs to be improved. However, the SIRS model, which evolved from the SIR model and has a more comprehensive design, has gradually been applied to the study of risk contagion and the domain of complex financial networks, including Chen et al. [40]. Yang, Qian et al. [41] constructed an SIR-SIR model to reveal the interaction between information dissemination and credit risk contagion.

Achieving carbon neutrality in business is a long-term transition process in which it is necessary to spread green and low-carbon concepts among companies, implement low-carbon environmental strategies and controlling the impact of the spread of misconceptions about high energy consumption. Based on the above considerations, this paper chooses to study energy enterprises in the financial market as its subject. We consider a micro-level scenario that takes into account the ability of individual affiliates to be infected with multiple ideas of high energy consumption and then use the SEIJRS infectious disease model, which includes latent and isolated states, to study the phenomenon of the cross infection of decisions about high energy consumption. This informs third-party energy consumption managers and government administrations in developing sound and optimal prevention and control strategies.

### 3. Methodology

*3.1. Basic Assumptions*

In the market system of energy companies, different types of companies trade closely with each other, and the phenomena of companies implementing high energy consumption strategies to increase their benefits due to market crowding, changes in related strategies and changes in market trends often spread between different companies. In order to fit the current situation of China's green economy context, this study combines the SEIJRS contagion model with the evolutionary characteristics of green and low-carbon belief transmission and proposes the following hypotheses:

**Hypothesis 1.** *Companies in a state of infection spread the concept of high emissions through their connections, leading to the infection of companies with which they have trade relationships, etc., and there is no fixed direction in the process of spreading the concept.*

**Hypothesis 2.** *Companies that have implemented high energy consumption strategies can be infected again.*

**Hypothesis 3.** *The whole market is a closed system which no relevant firms are allowed to exit and no foreign firms are allowed to join during the transmission of high emissions perceptions, i.e., the total number of firms in the market is fixed and the total number of individual firms is set to be a constant K.*

**Hypothesis 4.** *In an associated network of K firms, firms are classified into five states: susceptible (S), latent (E), infected (I), recovered (J) and healthy (R).*

(1) Susceptible state (S): Companies that are vulnerable to infection tend to be neutral in their own attitudes, but at the same time are vulnerable to the influence of associated companies with high energy consumption strategies, which can lead to their own perceptions being shaken into a latent state. In the market, vulnerable individuals are often institutions and companies with a low cash flow, high gearing or excessive financial leverage.

(2) Latent state (E): Companies in the latent state are shaken by the immediate benefits of high energy consumption because of the market squeeze they have received, but the current state is still internalised and it takes some time before the symptoms are revealed. At the same time, as their own perceptions increase, they also have an impact on the other companies involved, although less so than the individuals in the infected state. It is also because their own strategies have not yet been implemented that they can often take remedial measures to bring themselves into a healthy state. The specific signs of this type of business are a continuing decline in profits, rising debt ratios and a continuing increase in accounts receivable.

(3) Infection status (I): Infected companies are those that have already started to implement high energy consumption strategies and have a strong capacity to disseminate ideas. The change in the proportion of companies in this category in the market network

indicates the severity of the spread of high energy consumption strategies. It is difficult for the company to recover on its own and often requires the assistance of external forces to enter a state of recovery. This is evident in the market for companies and financial institutions that have serious problems such as severe cash flow shortfalls, insolvency and long-term losses.

(4) Rehabilitation status (J): Individual enterprises enter a state of recovery when their own low-carbon beliefs increase after receiving external controls or remediation, while their own ability to spread misconceptions due to external regulation is rapidly reduced. After the rehabilitation phase, these companies' own low-carbon beliefs and their control capabilities are effectively enhanced and they enter a healthy state and begin to gradually implement energy consumption control strategies.

(5) Health status (R): A healthy company is one that is well equipped with green and low-carbon technologies and has a strong commitment to green strategies. However, a healthy status does not mean permanent immunity to high energy consumption policies, and some healthy individuals lose their immunity and become susceptible to infection when the internal and external environment changes. Examples of healthy individuals are companies and organisations with adequate cash flow, low gearing and low leverage.

*3.2. Model Building*

According to Hypothesis 3 and Hypothesis 4, it can be seen that:

$$S(t) + E(t) + I(t) + J(t) + R(t) = K$$

According to the change in each sub-market node in the contagion of high energy consumption perceptions, the contagion process of high energy consumption perceptions with a constant total number of nodes is shown in Figure 1.

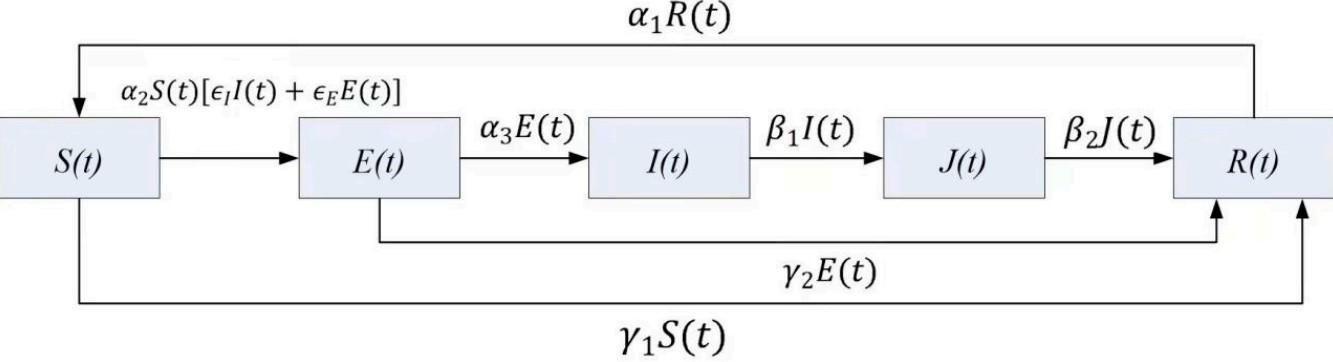

**Figure 1.** Cross-contamination process of corporate high energy consumption perceptions.

The SEIJRS infectious disease model was constructed based on the implementation of the perception contagion process of high energy consumption strategy among associated enterprises; the model consists of a set of differential equations (Equation (1)). The setting and meaning of each parameter in the model are shown in Table 1.

$$\begin{cases} \dot{S} = \alpha_1 R(t) - \alpha_2 S(t)[\epsilon_I I(t) + \epsilon_E E(t)] - \gamma_1 S(t) \\ \dot{E} = \alpha_2 S(t)[\epsilon_I I(t) + \epsilon_E E(t)] - \alpha_3 E(t) - \gamma_2 E(t) \\ \qquad \dot{I} = \alpha_3 E(t) - \beta_1 I(t) \\ \qquad \dot{J} = \beta_1 I(t) - \beta_2 J(t) \\ \quad \dot{R} = \gamma_1 S(t) + \gamma_2 E(t) + \beta_2 J(t) - \alpha_1 R(t) \end{cases} \tag{1}$$

**Table 1.** Definition of parameters for a model of cross-contamination of highly contaminated ideas between associated enterprises.

| Parameters | Definition |
|:---:|:---:|
| $\alpha_1$ | Probability of a business returning from a healthy state to a susceptible state |
| $\alpha_2$ | Infection rate |
| $\alpha_3$ | The probability of a business being infected in a latent state |
| $\epsilon_I$ | Correction factor relating to transmission from latent-state enterprises |
| $\epsilon_E$ | Correction factor related to transmission from infected-state enterprises |
| $\beta_1$ | The probability of an infected company being controlled |
| $\beta_2$ | The probability of an infected company being infected |
| $\gamma_1$ | Direct immunisation rate |
| $\gamma_2$ | Self-cure rate |

*3.3. Model Analysis*

**Theorem 1.** *The equilibrium point for the cross-contamination model of high energy consumption strategies among affiliated firms is:* $(\frac{\alpha_1 K}{\alpha_1+\gamma_1}, 0, 0, 0, \frac{\gamma_1 K}{\alpha_1+\gamma_1})$ *or* $(S_2, E_2, I_2, J_2, R_2)$, *among them:*

$$S_2 = \frac{k_1(\gamma_2+\alpha_3)}{\alpha_2(\epsilon_I k_1 + \epsilon_E k_2)}, \ k_1 = \frac{\beta_1}{\alpha_3}, \ k_2 = \frac{\beta_1}{\beta_2}$$

$$E_2 = k_1 I_2, \quad J_2 = k_2 I_2, \quad R_2 = K - S_2 - E_2 - I_2 - J_2$$

$$I_2 = \frac{K\alpha_1\alpha_2(\epsilon_I k_1 + \epsilon_E k_2) - k_1(\alpha_1+\gamma_1)(\gamma_2+\alpha_3)}{\alpha_2(\epsilon_I k_1 + \epsilon_E k_2)[\alpha_1(k_1+k_2)+k_1(\gamma_2+\alpha_3)]}$$

**Proof.** $R = K - S - E - I - J$, therefore, the cross-infection model of high energy consumption strategy among related companies can be converted into:

$$\text{Prove}: \begin{cases} \dot{S} = \alpha_1(K-S-E-I-J) - \alpha_2 S(t)[\epsilon_I I + \epsilon_E E] - \gamma_1 S \\ \dot{E} = \alpha_2 S[\epsilon_I I + \epsilon_E E] - \alpha_3 E(t) - \gamma_2 E \\ \dot{I} = \alpha_3 E - \beta_1 I \\ \dot{J} = \beta_1 I - \beta_2 J \end{cases} \quad (2)$$

When the model reaches the equilibrium point, $\dot{S} = \dot{E} = \dot{I} = \dot{J} = 0$, substitute into Equation (2):

$$\begin{cases} \alpha_1(K-S-E-I-J) - \alpha_2 S(t)[\epsilon_I I + \epsilon_E E] - \gamma_1 S = 0 \\ \alpha_2 S[\epsilon_I I + \epsilon_E E] - \alpha_3 E(t) - \gamma_2 E = 0 \\ \alpha_3 E - \beta_1 I = 0 \\ \beta_1 I - \beta_2 J = 0 \end{cases} \quad (3)$$

It is easy to know from Equation (3) that $E = \frac{\beta_1}{\alpha_3} I$, $J = \frac{\beta_1}{\beta_2} I$; we set: $k_1 = \frac{\beta_1}{\alpha_3}$, $k_2 = \frac{\beta_1}{\beta_2}$, and substitute it into Equation (3) to obtain:

$$I[\alpha_2 S(\epsilon_I + \epsilon_E k_1) - k_1(\alpha_3 + \gamma_2)] = 0$$

The following is discussed according to the following situation: When $I = 0$, $E = J = 0$, replace it back to Equation (3) and we can obtain $S = \frac{\alpha_1 K}{\alpha_1+\gamma_1}$, $R = \frac{\gamma_1 K}{\alpha_1+\gamma_1}$. When $I \neq 0$, $S = \frac{k_1(\gamma_2+\alpha_3)}{\alpha_2(\epsilon_I k_1 + \epsilon_E k_2)}$, then substitute the obtained S E J into Equation (3) to obtain:

$$I_2 = \frac{K\alpha_1\alpha_2(\epsilon_I k_1 + \epsilon_E k_2) - k_1(\alpha_1+\gamma_1)(\gamma_2+\alpha_3)}{\alpha_2(\epsilon_I k_1 + \epsilon_E k_2)[\alpha_1(k_1+k_2)+k_1(\gamma_2+\alpha_3)]}$$

To sum up, the two sets of solutions of Equation (2) are:

$$\begin{cases} S_1 = \dfrac{\alpha_1 K}{\alpha_1 + \gamma_1} \\ E_1 = 0 \\ I_1 = 0 \\ J_1 = 0 \end{cases} \text{and} \quad \begin{cases} S_2 = \dfrac{k_1(\gamma_2 + \alpha_3)}{\alpha_2(\epsilon_I k_1 + \epsilon_E k_2)} \\ E_2 = k_1 I_2 \\ J_2 = k_2 I_2 \\ I_2 = \dfrac{K\alpha_1\alpha_2(\epsilon_I k_1 + \epsilon_E k_2) - k_1(\alpha_1 + \gamma_1)(\gamma_2 + \alpha_3)}{\alpha_2(\epsilon_I k_1 + \epsilon_E k_2)[\alpha_1(k_1 + k_2) + k_1(\gamma_2 + \alpha_3)]} \end{cases} \text{, and the theorem can}$$

be proved. □

By observing the two equilibrium points obtained, it can be found that, the equilibrium point $(\frac{\alpha_1 K}{\alpha_1 + \gamma_1}, 0, 0, 0, \frac{\gamma_1 K}{\alpha_1 + \gamma_1})$ is actually the balance point of the spread of the system's unrelated high energy consumption strategy. According to the condition of Hypothesis 2, individuals who have passed the associated high energy consumption strategy can be infected again. Therefore, if there is no systematic control in the model, it is impossible for the high energy consumption strategy to have an equilibrium point of risk-free propagation in the process of propagation.

The optimal control problem is solved in order to minimise the sum of the number of firms in latent, infected and recovered states and the cost of government bailout measures over a fixed period of time.

## 4. The Optimal Control Problem

For third-party energy consumption managers, the objective is to control the spread of the concept over a period of time in order to reduce the number of companies affected by the concept (the categories of companies affected are latent, infected and recovered), while for those already affected by the concept of high energy consumption, the main measures to be taken by the government are to provide assistance and rehabilitation to those companies already infected. Taken together, the overall objective of the government sector is to minimise the sum of the number of individuals in latent, infected and recovered states and the cost of relief measures over a fixed period of time. Therefore, within, the objective function is:

$$\min_{\beta_1(t)} Y = \int_0^T \left[ m_1 E(t) + m_2 I(t) + m_3 J(t) + \frac{\mu \beta_1^2(t)}{2} \right] dt$$

where $m_1$, $m_2$, $m_3$ are the weights of the latent state, infection state and rehabilitation state, and $\frac{\mu \beta_1^2(t)}{2}$ represents the individual's rescue costs. It is assumed that the allowed control sets are:

$$W = \{\beta_1(t) \in L(0, T) | \beta_{min}(t) \leq \beta_1(t) \leq \beta_{max}(t)\}$$

$R = K - S - E - I - J$, so, in the optimal control problem, it is not regarded as a state variable. The optimal control problem with restricted control is defined as follows:

$$\min_{\beta_1(t)} Y = \int_0^T \left[ m_1 E(t) + m_2 I(t) + m_3 J(t) + \frac{\mu \beta_1^2(t)}{2} \right] dt$$

$$\text{s.t.} \quad \dot{S} = \alpha_1(K - S - E - I - J) - \alpha_2 S(t)[\epsilon_I I + \epsilon_E E] - \gamma_1 S$$

$$\dot{E} = \alpha_2 S[\epsilon_I I + \epsilon_E E] - \alpha_3 E(t) - \gamma_2 E$$

$$\dot{I} = \alpha_3 E - \beta_1 I \quad \dot{J} = \beta_1 I - \beta_2 J$$

$$S(0) = S_0 \quad E(0) = E_0 \quad I(0) = I_0$$

$$J(0) = J_0 \quad \beta_{min}(t) \leq \beta_1(t) \leq \beta_{max}(t)$$

The optimal control problem is solved below:

After observing the form, it can be seen that it is a Lagrangian problem. The state equation is written in the form of a constraint equation:

$$\alpha_1(K - S - E - I - J) - \alpha_2 S(t)[\epsilon_I I + \epsilon_E E] - \gamma_1 S - \dot{S} = 0 \tag{4}$$

$$\alpha_2 S[\epsilon_I I + \epsilon_E E] - \alpha_3 E(t) - \gamma_2 E - \dot{E} = 0 \tag{5}$$

$$\alpha_3 E - \beta_1 I - \dot{I} = 0 \tag{6}$$

$$\beta_1 I - \beta_2 J - \dot{J} = 0 \tag{7}$$

The augmented functional is constructed by the Lagrange multiplier method:

$$Y' = \int_0^T \left\{ \left[ m_1 E(t) + m_2 I(t) + m_3 J(t) + \frac{\mu \beta_1^2(t)}{2} \right] + \lambda_1(t)[\alpha_1(K - S - E - I - J) - \alpha_2 S(t)[\epsilon_I I + \epsilon_E E] - \gamma_1 S - \dot{S}] \right. \\ \left. + \lambda_2(t)[\alpha_2 S[\epsilon_I I + \epsilon_E E] - \alpha_3 E(t) - \gamma_2 E - \dot{E}] + \lambda_3(t)[\alpha_3 E - \beta_1 I - \dot{I}] + \lambda_4(t)[\beta_1 I - \beta_2 J - \dot{J}] \right\} dt \tag{8}$$

where $\lambda(t)$ are the undetermined adjoint vectors. Define the Hamiltonian function:

$$H[S, E, I, J, R, \beta_1, \lambda_1, \lambda_2, \lambda_3, t] = m_1 E(t) + m_2 I(t) + m_3 J(t) + \frac{\mu \beta_1^2(t)}{2} + \lambda_1 \dot{S} + \lambda_2 \dot{E} + \lambda_3 \dot{I} + \lambda_4 \dot{J}$$

Then, the augmented functional can be reduced to:

$$Y' = \int_0^T \left\{ H[S, E, I, J, R, \beta_1, \lambda_1, \lambda_2, \lambda_3, t] - \lambda_1 \dot{S} - \lambda_2 \dot{E} - \lambda_3 \dot{I} - \lambda_4 \dot{J} \right\} dt$$

When $Y'$ reaches the extreme value, the adjoint vector of the whole dynamic system needs to satisfy the adjoint equation, which is:

$$\begin{cases} \dot{\lambda}_1 = -\frac{\partial H}{\partial S} = \lambda_1(\alpha_1 + \gamma_1) + \alpha_2(\lambda_1 - \lambda_2)(\epsilon_I I + \epsilon_E E) \\ \dot{\lambda}_2 = -\frac{\partial H}{\partial E} = -m_1 + \lambda_1 \alpha_1 + (\lambda_1 - \lambda_2)\alpha_2 S \epsilon_E + (\lambda_2 - \lambda_3)\alpha_3 - \lambda_2 \gamma_2 \\ \dot{\lambda}_3 = -\frac{\partial H}{\partial I} = -m_2 + \lambda_1 \alpha_1 + (\lambda_1 - \lambda_2)\alpha_2 S \epsilon_I + (\lambda_3 - \lambda_4)\beta_1 \\ \dot{\lambda}_4 = -\frac{\partial H}{\partial J} = -m_3 + \lambda_1 \alpha_1 + \lambda_4 \beta_2 \end{cases}$$

At the same time, the control variable $\beta_1$ meets the control equation:

$$\frac{\partial H}{\partial \beta_1} = 0$$

By solving the governing equation, we can obtain:

$$\beta_1^* = \min\left( \max\left( \beta_{\min}, \frac{(\lambda_3 - \lambda_4)I}{\mu} \right), \beta_{\max} \right)$$

## 5. The Numerical Simulation

Due to the large number of differential equations and parameters, it is difficult to find analytical solutions, so this section is devoted to the numerical simulation of the system of differential equations obtained in Section 3.

Assume that the initial values in each state are: $S(0) = 0.95K, E(0) = 0.02K$, $I(0) = 0.01K, J(0) = 0.01K, R(0) = 0.01K$. Among them, K is the total number of companies in the market. The value range of the rescue rate $\beta_1$ and early recovery rate $\gamma_2$ is $[0, 0.9]$. The time frame of the study was $[0, 100]$. Other parameter values in the optimal control problem are shown in Table 2.

**Table 2.** Values of various parameters of the optimal control model.

| Parameter | Parameter Value | Parameter | Parameter Value | Parameter | Parameter Value | Parameter | Parameter Value |
|---|---|---|---|---|---|---|---|
| $\alpha_1$ | 0.1 | $m_1$ | 0.5 | $\beta_2$ | 0.05 | $\epsilon_I$ | 0.8 |
| $\alpha_2$ | 0.1 | $m_2$ | 0.8 | $\gamma_1$ | 0.1 | $\epsilon_E$ | 0.2 |
| $\alpha_3$ | 0.1 | $m_3$ | 0.4 | $\mu_1$ | 6 | $\mu_2$ | 3 |

In this paper, Matlab R2018a is used as the numerical simulation platform, and Ode45 is used as the solver to solve the adjoint equation, control equation and control condition after simultaneous establishment.

Figure 2 depicts the dynamics of the number of energy consumption enterprises, the optimal self-healing rate, the optimal control rate and the number of control regenerations for each state. The optimal self-healing rate decreases over time, while the optimal control increases first to a peak and then decreases. The control regeneration number in the initial phase is greater than one. The implementation of the optimal control strategy causes it to gradually drop below one over time and the spread of high energy consumption perceptions is controlled. The control regeneration numbers in the latter part of the study phase returned to greater than one, due to a lax mentality on the part of individual companies and government control authorities during this period, and the rapid decline in self-cure and control rates led to an increase in the spread of high energy consumption ideas. We found that the number of companies in the susceptible and latent states decreased over time to zero. The number of companies in the infected state increased to a peak and then decreased to 0.007 K. The number of companies in the controlled state increased slightly to a peak and then decreased gradually to zero. The number of companies in the healthy state increased over time and finally increased to 0.98 K before levelling off.

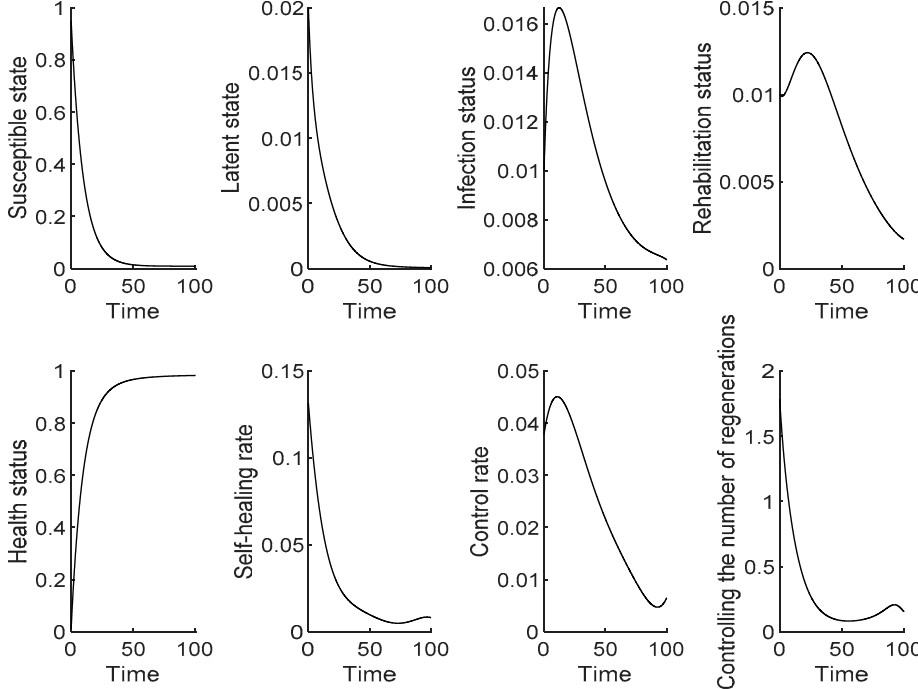

**Figure 2.** Dynamics of the number of firms, the optimal control variables and the number of control regenerations in each state.

## 5.1. Influence of Contagion Rate on Optimal Control Strategies

The evolution of the number of firms, the optimal self-healing rate, the optimal control rate and the number of control regenerations in each state when the contagion rate is taken to be 0.1, 0.3 and 0.9, respectively, is shown in Figure 3. We find that the number

of enterprises in the susceptible state decreases faster as the infection rate increases. In contrast, the number of enterprises in the healthy state decreases slightly as the infection rate increases. The number of enterprises in the remaining three states increases as the infection rate increases. As the contagion rate of high energy consumption perceptions increases, the process rate of change in the number of firms in each state of optimal self-healing at different contagion rates and the rate of optimal control increase. It is important to note that the trend of the self-healing rate increases to a peak and then decreases to zero when the transmission rate achieves a large value of 0.9. We found that the rate of self-cure decreased at a slower rate than the rate of control. In the early stages of transmission of high energy consumption ideas, the cure rate of the enterprises themselves is greater than the control rate of the government departments taking control measures on vulnerable enterprises, and the control rate in the later stages is greater than the cure rate.

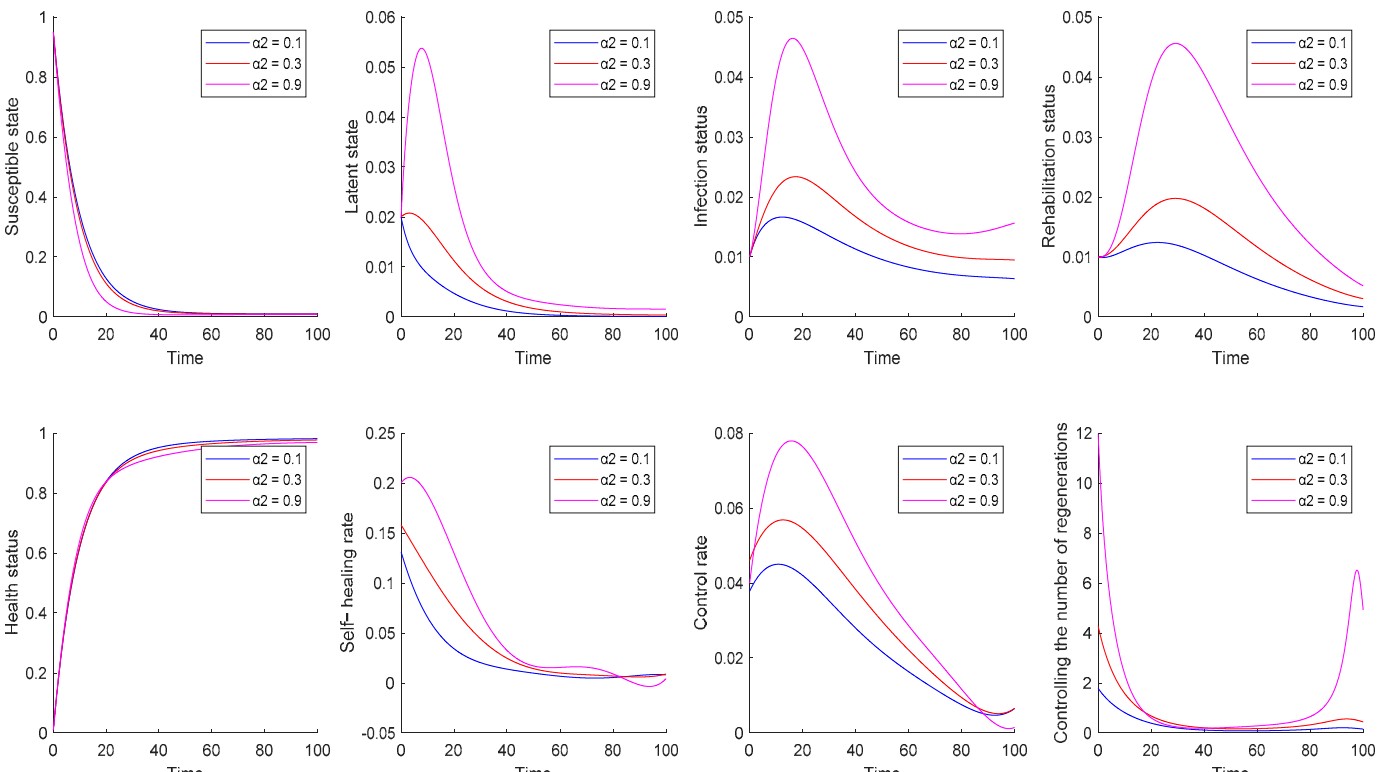

**Figure 3.** The dynamics of the number of firms in each state, the optimal control variables and the number of control regenerations at different rates of contagion.

We therefore suggest that, when the contagion rate of high energy consumption concepts increases, senior managers of enterprises in the latent stage should promptly formulate reasonable countermeasures, establish green and low-carbon environmental awareness and actively respond to the national call to avoid being infected with high energy consumption concepts. At this time, the government control department should strengthen the control of the already infected enterprises and help them to establish green and low-carbon awareness, implement environment-friendly strategies and achieve green transformation by strengthening the control of vulnerable enterprises to reduce the scope of the transmission of high energy consumption concepts, and implementing both incentives and penalties for them. The number of control regenerations increases as the transmission rate increases, with the trend being to first fall below one, and the number of control regenerations rising sharply again at higher transmission rates, making control more difficult again. It shows that, with the increased rate of transmission and the fact that companies can be infected multiple times, government departments and companies in latent

states must always develop optimal control strategies based on the form of transmission of highly contaminated concepts and must not let down their guard and reduce their control efforts because of a brief improvement in form.

### 5.2. The Impact of Infection Rates in Latent-State Firms on Optimal Control Strategies

When the probability of a company in the latent state being infected by high energy consumption perceptions changes, the changes in the number of companies in each state, the optimal control strategy and the number of control regenerations are shown in Figure 4. When increasing, it means that the decision makers of the enterprises in the latent state do not realise their problems and self-control in time, the self-healing rate is low and the probability of being infected by high energy consumption perceptions increases. This leads to a large number of companies in this state being infected; thus, the number of companies in the infected state increases and the number of submarkets in the latent state decreases. At this point, government departments take control measures to control the output of highly polluting ideas from infected companies by increasing the rate of control and narrowing the spread of highly polluting ideas, thus leading to an increase in the number of companies in a controlled state. We found that the number of control regenerations increased with the increase and that the trend was roughly the same as at the time of the change. The number of enterprises in the susceptible and recovered states was barely affected by the increase.

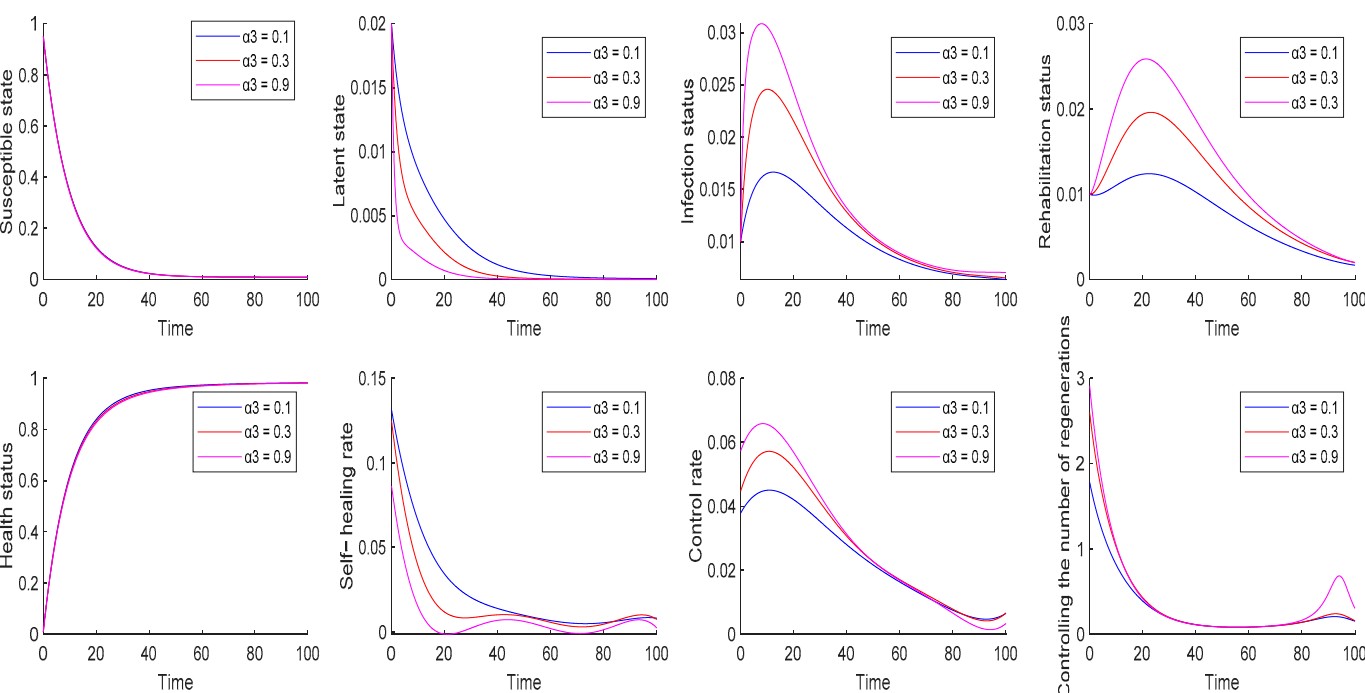

**Figure 4.** Variation in the number of firms in each state, the optimal control strategy and the number of control regenerations under different $\alpha_3$.

### 5.3. The Effect of Direct Immunity Rates on Optimal Control Strategies

The number of enterprises in each state, the evolution of the optimal self-cure and control rates and the number of control regenerations at different direct immunisation rates are shown in Figure 5. An increase in the direct immunity rate means that companies with a strong low-carbon mindset and green beliefs are able to control high energy consumption strategies in a shorter period of time in the event of a major future event impacting on their green beliefs.

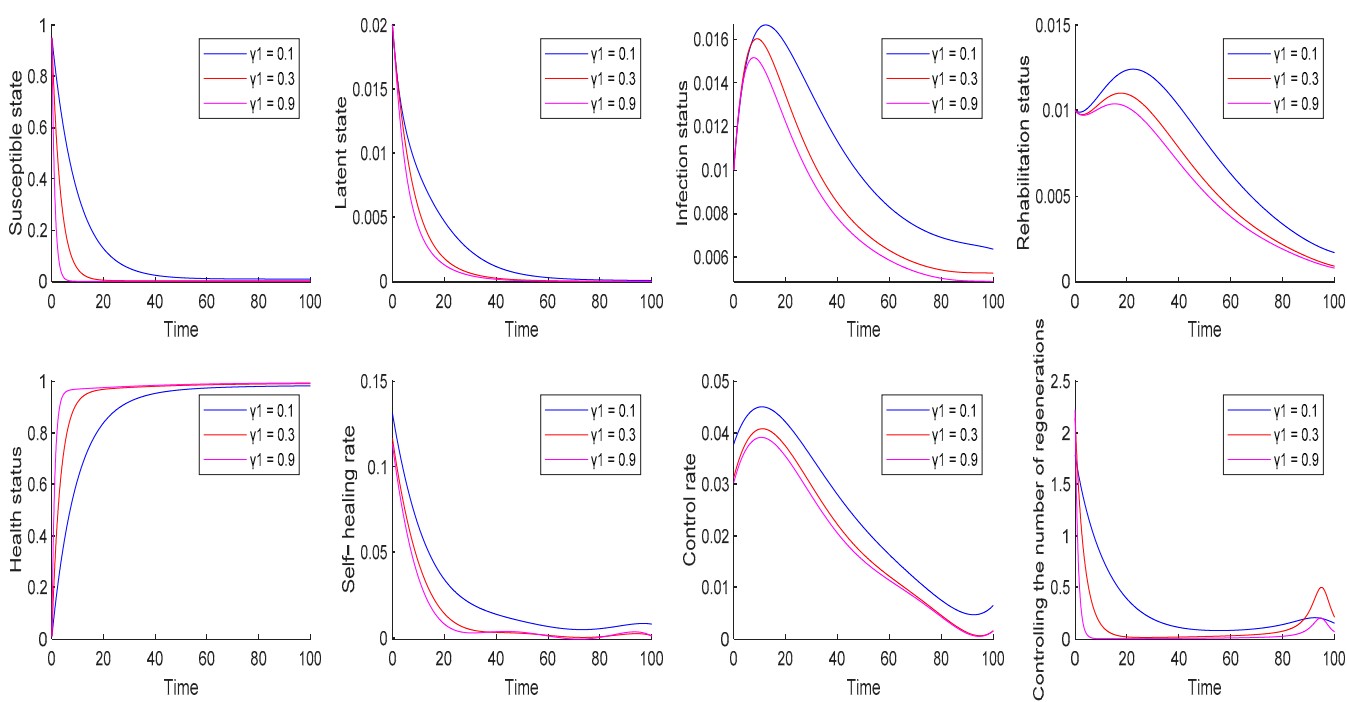

**Figure 5.** Evolution of the number of firms, the optimal control variables and the number of control regenerations in different $\gamma_1$ states.

It is therefore a priority for companies to raise their own immunity rate, to spread green concepts within the company, to establish low-carbon beliefs, to strengthen environmental awareness and to improve their defenses when high energy consumption concepts start to spread. When companies in the whole market system generally have a higher awareness of green concepts, they have a stronger ability to resist the spread of high energy consumption concepts and can control the spread of highly polluting concepts in a shorter period of time, preventing further expansion of environmental pollution.

## 6. Conclusions and Limitations

This paper combines and improves the medical infectious disease model and finally chooses to use the SEIJRS infectious disease model, which includes the latent and isolated states. The model is used to study the phenomenon of the cross-contamination of high energy consumption perceptions in energy enterprises. Through an in-depth study of the various types of firms during the period affected by the perception of high energy consumption, we classified the individuals associated in the study network into five states: susceptible, latent, infected, recovered and healthy states. The dynamics of the number of firms, the optimal control variables and the number of control regenerations in different states were obtained by varying different parameters, and the results are presented in Table 3.

**Table 3.** Optimal control strategies for different scenarios.

| Situation | Optimal Control Strategy |
|---|---|
| Increased transmission rate | Increased self-cure rates and increased control rates |
| Increased infection rates in latent-state businesses | Decrease in self-cure rates and increase in control rates |
| Increased direct immunisation rates | Decrease in both self-cure and control rates |

When enterprises are influenced by the external environment and green low-carbon beliefs are not solid, the contagion rate of high energy consumption strategy increases;

senior management of enterprises in the latent state should promptly consolidate their own low-carbon environmental protection concept, abandon the idea of high energy consumption and adopt optimal prevention and control strategies in a timely manner to increase the self-healing rate of enterprises themselves, rather than wait for the intervention and control of government departments after they have started to adopt high energy consumption strategies and cause pollution to the environment.

For the government administration, it should increase the control of enterprises that have started to implement high energy consumption strategies, increase efforts to spread green ideas, impose penalties and controls on non-compliant enterprises, and reward good performance to narrow the spread of high energy consumption ideas. When the probability of a business being infected increases in the latent state, it indicates that it has a weak green mindset and lacks strong beliefs, leading to a decrease in self-healing rates. In the face of bad decisions by business leaders in a latent state, the government administration should take preventive and control measures to save the situation and increase the intensity of control over companies that have implemented high energy consumption strategies. When the direct infectious rate increases, the whole market system has a stronger belief in green strategies, and companies in a vulnerable state are able to recover quickly, stop implementing high energy consumption strategies and switch to implementing environmentally friendly strategies. The number of firms in the remaining states decreases as the direct immunity rate increases, and the self-healing rate of firms in the latent state and the rate of government control over infected firms both decrease as the direct immunity rate increases. Therefore, a more moderate optimal control strategy can be adopted at this point.

It Is important to note that, since the number of infected companies peaks at the same time as the balance between healthy and susceptible companies is reached, government authorities should not reduce prevention and control efforts at this stage because of the increase in the proportion of healthy companies, but should still focus on regulation and monitoring to ensure effective control of the energy consumption concept. The different nature of the various types of enterprises and their resistance to misconceptions during the influence of high energy consumption strategy is an aspect that is not taken into account in the model developed, so the model could be further improved to suit the reality.

**Author Contributions:** Conceptualization, S.W.; methodology, software, validation, formal analysis, and writing—original draft preparation, S.W.; investigation and resources, S.W. and W.C.; writing—review and editing, and supervision, W.C. and G.W. All authors have read and agreed to the published version of the manuscript.

**Funding:** This research received no external funding.

**Institutional Review Board Statement:** Not applicable.

**Informed Consent Statement:** Not applicable.

**Data Availability Statement:** The authors would like to thank the editors and anonymous reviewers for their thoughtful and constructive comments.

**Conflicts of Interest:** The authors declare no conflict of interest.

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
