# Peer review of "The Impact of the Low-Carbon Energy Concept and Green Transition on Corporate Behaviour—A Perspective Based on a Contagion Model"

_sustainability, doi:10.3390/su142416600_

Round 1

Reviewer 1 Report

1. In the model building section, why is the correction factor related to the transmission of state enterprises considered? Do other types of enterprises need to be considered as well?

2. Is the logical relationship of applying the improved SEIJRS infectious disease model to this problem consistent with the problem? There is less support in the literature in this regard in the paper.

3. How to determine the contagion rate of high energy consumption in the real business operation? Moreover, for supply chain enterprises, the increase in energy consumption value of upstream enterprises will definitely drive downstream industries.

Author Response

Thank you for your advice, all your suggestions are very important and they will guide me in my thesis writing and research work.

  1. This paper does not examine the relationship between state enterprises and the correction factor separately. This paper examines the diffusion of energy-intensive ideas in markets made up of energy-consuming firms, and includes all energy-consuming firms in the scope of the study.The different nature of the various types of enterprises and their resistance to misconceptions during the influence of high energy consumption strategy is an aspect that is not taken into account in the model developed, so the model could be further improved to suit the reality.
  2. The applicability of the SEIJRS model has been ensured by analysing the similarities between the spread of high energy consumption ideas and the spread of infectious diseases. At present, scholars have conducted fewer studies on the spread of high-energy consumption ideas using mathematical models, and contagion models have been mainly applied to aspects such as risk transmission in financial markets. In view of the similarity between financial risk transmission and idea transmission, this paper adopts an improved contagion model to study the transmission of high-energy consumption ideas.
  3. The rate of transmission of energy-intensive concepts in actual business operations is determined by a number of factors, including, but not limited to, the environmental beliefs of the decision makers, the speed with which the company takes self-help measures, the herd effect, and the economic situation of the company itself. During the spread of the concept of high energy consumption, different types of companies (upstream and downstream companies in the chain) lead to different rates of contagion and self-healing, which are not taken into account in this paper, so the model needs to be further improved. Government and third-party control authorities need to determine infection rates and make timely adjustments through survey analysis of companies in their networks.

Reviewer 2 Report

The authors used the improved SEIJRS infectious disease model to analyze the phenomenon of corporate high energy strategy infection, and they used optimal control theory to provide suggestions for governments and decision makers to develop reasonable optimal prevention and control strategies. The research is interesting, but there still exist some parts that are not clear or rigorous enough. The detailed comments are listed as the followings.

1.      In the introduction section, most of the sentences are without any citations, which includes lots of knowledge that is not found by this research (Other sections of this article have similar problems). It is a serious problem for an academic article. Please go through the whole paper carefully (especially the introduction part), and add citations to the corresponding sentences.

2.      Rewrite the first sentence of the introduction section. Here, “a common problem” is not clear enough. Please briefly state the problems caused by climate change, rewrite the sentence, and add citations.

3.      P2: Briefly clarify what is “lock in emissions”.

4.      2.1: Go through the paper and change “CO2” to “CO2”. Where you use CO2 for the first time in this paper, you should mention the full name, i.e., carbon dioxide (CO2).

5.      Add the full name of SEIJRS where you used it for the first time in this paper.

6.      2.4: Clearly state what is the advantages of the infection disease model compared with other kinds of methods (briefly list other widely used models together with their limitations)  and why you choose this method.

7.      In the Methodology section, it would be better to add a subchapter to introduce all the data you used in this research. Clearly state what dataset did you use, with detailed information such as the format of the data, time window, and the source of the data (such as doi, or references). Is there any data preprocess did you make in the experiments?

8.      3.2: It would be better if you could clearly and briefly clarify what is model input, and what is output.

9.      “Therefore, if there is no systematic control in the model, … in the process of propagation.\” Delete “\”.

10.   Figure2-5: Add subtitles such as (a), (b), (c)… to each subplot. The font size is too small.

Author Response

Thank you for your advice, all your suggestions are very important and they will guide me in my thesis writing and research work.

  1. I have added citations and amended the content.
  2. I have recorrected this sentence.
  3. “lock in emissions” is inappropriate and has been removed.
  4. All  have been corrected and mention the full name.
  5. The SEIJRS infectious disease model is an improved mathematical model of epidemic diseases, which is more in line with the spread of high energy consumption concepts.
  6. Comparisons of other models with this model have been added.
  7. The individual parameters in the numerical simulation are variable and are subject to change depending on the actual situation. The numerical simulation part investigates the dynamic change process under high energy consumption propagation .
  8. The model assumes that firms in the market form a closed system with no inputs or outputs, that the number of firms in the system is constant, and that only firms in different states switch between each other.
  9. Already removed "\" .
  10. The image has been enlarged.

Round 2

Reviewer 2 Report

Most of the comments are answered suitably, I suggest accepting the manuscript.